# Performance of Waist-To-Height Ratio, Waist Circumference, and Body Mass Index in Discriminating Cardio-Metabolic Risk Factors in a Sample of School-Aged Mexican Children

**DOI:** 10.3390/nu10121850

**Published:** 2018-12-01

**Authors:** Ibiza Aguilar-Morales, Eloisa Colin-Ramirez, Susana Rivera-Mancía, Maite Vallejo, Clara Vázquez-Antona

**Affiliations:** 1Department of Social Medicine Research, National Institute of Cardiology ‘Ignacio Chávez’, Juan Badiano 1, Sección XVI, Mexico City C.P. 14080, Mexico; taniaibizaam@gmail.com (I.A.-M.); syriverama@conacyt.mx or yesqfb@yahoo.com.mx (S.R.-M.); maite_vallejo@yahoo.com.mx (M.V.); 2CONACYT—National Institute of Cardiology ‘Ignacio Chávez’, Juan Badiano 1, Sección XVI, Mexico City C.P. 14080, Mexico; 3Department of Pediatric Echocardiography, National Institute of Cardiology ‘Ignacio Chávez’, Juan Badiano 1, Sección XVI, Mexico City C.P. 14080, Mexico; cvazquezant@yahoo.com.mx

**Keywords:** waist-to-height ratio, children, obesity, body mass index, waist circumference, cardio-metabolic risk

## Abstract

The most common tools used to screen for abdominal obesity are waist circumference (WC) and waist-to-height ratio (WHtR); the latter may represent a more suitable tool for the general non-professional population. The objective of this study was to evaluate the association of WHtR, WC, and body mass index with lipidic and non-lipidic cardio-metabolic risk factors and the prediction capability of each adiposity indicator in a sample of school-aged Mexican children. Overall, 125 children aged 6 to 12 years were analyzed. Anthropometric, biochemical, and dietary parameters were assessed. Receiving operating characteristic (ROC) analysis and univariate and multivariate linear and logistic regression analyses were performed. All the three adiposity indicators showed significant areas under the ROC curve (AURC) greater than 0.68 for high low-density lipoprotein cholesterol (LDL-c), triglycerides, and atherogenic index of plasma, and low high-density lipoprotein cholesterol (HDL-c). A significant increased risk of having LDL-c ≥ 3.4 mmol/L was observed among children with WHtR ≥ 0.5 as compared to those with WHtR < 0.5 (odds ratio, OR: 2.82; 95% confidence interval, CI: 0.75–7.68; *p* = 0.003). Fasting plasma glucose was not associated with any of the adiposity parameters. WHtR performed similarly to WC and z-BMI in predicting lipidic cardio-metabolic risk factors; however, a WHtR ≥ 0.5 was superior in detecting an increased risk of elevated LDL-c.

## 1. Introduction

Childhood obesity remains a major public health problem worldwide [1], mainly affecting children in low- and middle-income countries [2]. In Mexico, in 2016, 33.2% of children aged between 5 and 11 years were overweight or obese [3]. Increased cardiovascular risk and the early-onset of chronic diseases are among the health consequences of childhood obesity [4,5]. It has been recognized that evaluating weight status by traditional tools such as body mass index (BMI) to screen for cardio-metabolic risks is limited by the fact that BMI does not reflect fat distribution [6]; for instance, accumulation of visceral fat (intra-abdominal obesity), compared to subcutaneous fat (peripheral), has been associated with a greater cardiovascular risk [7,8,9,10] and metabolic disorders such as high blood pressure, dyslipidemia, and altered glucose metabolism [11,12].

The most common tools used to screen for abdominal obesity are waist circumference (WC) and waist-to-height ratio (WHtR); nonetheless, it has been reported that people with equal WC but distinct height have different cardiovascular risk [13,14], highlighting the relevance of the WHtR, which incorporates height in its calculation. A WHtR ≥ 0.5 has been proposed as an indicator of abdominal obesity in both adults and children at any age [4,15] and it has been associated with a greater cardiovascular risk and proinflammatory response in pediatric populations [12,16,17,18]. Additionally, WHtR has the advantage of not being dependent on age or sex-specific percentiles relative to a reference population when used in pediatric population, as BMI or WC are; therefore, it may represent a more practical tool to evaluate abdominal obesity for the general non-professional population [19,20,21], which might have relevant public health implications.

Despite the utility of WHtR to screen for cardio-metabolic risk factors in children and its practical advantages in measurement and interpretation, its use is still limited to research purposes without a widespread use in the clinical and population settings. In Mexico, few studies have evaluated the utility of this index as a predictor of cardio-metabolic risk factors compared to WC and/or BMI in pediatric population [22,23,24]; additionally, to our knowledge, there is no study comparing the performance of these three methods in detecting cardio-metabolic risks factors accounting for relevant dietary indicators that may influence this association and contribute to the inconsistences observed regarding the superiority of WHtR over WC in diverse populations [21]. The aim of this study was to evaluate the association of WHtR and WC as indicators of abdominal adiposity, and BMI as an indicator of general adiposity, with lipidic and non-lipidic cardio-metabolic risk factors and their prediction capability, in a sample of school-aged Mexican children, considering the effect of key dietary indicators.

## 2. Materials and Methods

### 2.1. Study Population

This cross-sectional analysis included 125 healthy children aged between 6 to 12 years enrolled in a study aimed to evaluate echocardiography abnormalities associated with obesity in school age children. Participants were recruited from a public elementary school in Mexico City between July 2015 and February 2017 thorough informative sessions delivered by research team members to the parents of children in all grades to explain the study objectives and procedures involved; parents interested in the study were provided with further detailed information. Children were included if they and their parents or guardians provided written informed consent to take part of the study. Children who were underweight and/or had cardiovascular disease, hypertension, diabetes, or any other chronic condition were excluded from the study. For the purpose of this cross-sectional analysis, children with incomplete anthropometric or biochemical data were also excluded from the analysis. The study was approved by the Research and Ethics Committees of the National Institute of Cardiology Ignacio Chávez (Instituto Nacional de Cardiología Ignacio Chávez (INCICh)) (REF. PT-15-009).

### 2.2. Assessments

Potential participants identified at school were scheduled for a clinical visit at the INCICh to confirm eligibility and undergo clinical and dietary assessments.

#### 2.2.1. Anthropometry and Adiposity Indicators Definition

Weight, height and waist circumference were measured following standardized procedures described by the International Society for the Advancement of Kinanthropometry (ISAK) [25]. All anthropometric measures were taken by the same previously standardized research personnel. Standardization procedures were carried out according to the Habicht method [26].

Participants were asked to use light clothing without shoes. Height was measured to the nearest 0.1 cm using a Seca 220 stadiometer (seca GmbH & Co. KG., Hamburg, Germany). Children were barefoot and advised to stand in an upright position, with shoulders and arms relaxed and head in the Frankfort horizontal plane. Weight was determined to the nearest 0.1 kg using a calibrated Seca 700 mechanical column scale. Waist circumference (WC) was measured with a measuring tape made of glass fiber BodyFlex, with length of 150 cm and precision of 1 mm, at the mid-point between the iliac crest and the lower edge of the ribs, with the subject in a relaxed standing position and the tape situated in a horizontal plane directly on the skin after a normal exhalation.

BMI z-score was calculated, as an indicator of general adiposity, and classified according to the World Health Organization (WHO) age- and gender-specific growth standards with the WHO Anthro Plus software [27]. Normal weight was defined as a BMI z-score between −2 standard deviations (SD) and 1 SD; overweight as z-score > 1 SD; and obesity as z-score > 2 DE [28]. Abdominal obesity was defined by means of WC and WHtR. A WC ≥ 90 percentile for gender, age and height, specific for Mexican children [29], was considered to indicate abdominal obesity. WHtR was calculated by dividing WC by height in cm. A WHtR ≥ 0.5 was considered as an indicator of abdominal obesity [4].

#### 2.2.2. Biochemical Analysis and Cardio-Metabolic Risk Factors

Fasting blood samples (16 mL) were collected. For the purpose of the biochemical determinations employed in this sub-analysis, 3.5 mL were processed at the central laboratory of the INCICh to determine serum levels of glucose, triglycerides (TGs), low-density lipoprotein cholesterol (LDL-c), high-density lipoprotein cholesterol (HDL-c), and total cholesterol (TC). Elevated fasting plasma glucose (FPG) was defined according to the American Diabetes Association criteria as FPG ≥ 5.6 mmol/L [30]. Serum lipid levels were classified according to the criteria endorsed by the Expert Panel on Integrated Guidelines for Cardiovascular Health and Risk Reduction in Children and Adolescents [31] as follows: high TC, ≥5.2 mmol/L; high LDL-c, ≥3.4 mmol/L; high TG, ≥1.1 mmol/L for 0–9 years and ≥1.5 mmol/L for 10–19 years; low HDL-c, <1 mmol/L.

Additionally, two atherogenic indexes were calculated. The atherogenic index of plasma (AIP) was calculated as Log10 (TG/HDL-c), where a value <0.11 has been suggested as an indicator of low risk, 0.11–0.21 for intermediate risk and >0.21 for increased risk [32]. The atherogenic index (AI) was calculated as LDL-c/HDL-c; an elevated AI was defined as a value >3 for women and >2.5 for men [33].

#### 2.2.3. Dietary Intake

It was assessed with a multiple-pass 24-h food recall with parental assistance taken by a trained research dietitian [34]. Children and parents were asked to provide detailed information on the food and beverages consumed by the children the day before. Three-dimensional food models were used to assist in defining food type and portion sizes. Information collected in the 24-h food recall was analyzed by trained personnel with a nutrient software program (ESHA Food Processor SQL v.11.4.0; ESHA Research, Salem, OR, USA). Additional food items were added to the ESHA database when needed to account for local foods and better reflect the actual food consumed by the children. Intake of energy and macronutrients (protein, carbohydrates and lipids) was estimated.

### 2.3. Statistical Analysis

Descriptive statistics were expressed as mean ± standard deviation (SD) or median with 25th and 75th percentiles for continuous variables, depending on whether or not variables were normally distributed as evaluated by the Kolmogorov–Smirnoff test. Comparison of variables between boys and girls was performed by using the Student’s *t*-test or Mann–Whitney U test, as applicable. The receiving operating characteristic (ROC) analysis was employed to test the ability of all adiposity indicators (z-BMI, WC and WHtR) to discriminate children with cardio-metabolic risk factors from those with normal values through the areas under the ROC curve (AURC), for which 95% confidence intervals (CIs) were constructed. ROC curves for each cardio-metabolic risk factor were compared among the three adiposity indicators.

Univariate and multivariate linear regression analyses were performed to test the linear association between each adiposity indicator with each cardio-metabolic risk marker. Regression coefficients (β), 95% CI for β and standardized β were estimated for all linear models. In the case of WHtR, it was introduced in the linear models as the quotient of WHtR divided by 10, so the β indicated the increase in each cardio-metabolic risk marker per increase of 0.1 units in WHtR.

Binary logistic regression analyses were used to evaluate the influence of excess general (z-BMI > 1 SD) and abdominal (WC ≥ 90 percentile or WHtR ≥ 0.5) adiposity on each cardio-metabolic risk marker as dichotomous variables based on previously described cut-off points. Odds ratios (OR) and 95% CI were estimated for all logistic models. Both liner and binary logistic multivariate models were adjusted for age, sex and dietary variables, as applicable. Additionally, the Firth logistic regression method was applied to test the associations of abdominal adiposity by WHtR (≥0.5) with high LDL-c and by WC (≥90 percentile) with high AIP, due to quasi-complete separation of the data that led to an infinite maximum likelihood estimate for these associations [35,36]. These two multivariate Firth logistic regression models were adjusted for the same covariates.

All analyses were performed using SPSS Version 23.0 (SPSS, Inc., Chicago, IL, USA), except for the comparison of the AURC among the three adiposity indicators, which was performed applying the roccomp command in Stata v.14. For the Firth logistic regression, the IBM SPSS Statistics—Integration Plug-in for R was used. A *p* value < 0.05 was considered as significant.

## 3. Results

A total of 142 subjects were enrolled in the study, 17 of them had missing data; therefore, 125 were included in the analysis. The median (25th–75th percentile) age was 9 (8–10) years old, and 58% were girls. Anthropometric, biochemical and dietary intake variables stratified by gender are shown in Table 1. There were no significant differences between genders except for energy intake, that was higher in boys than girls (2081 ± 692.6 vs. 1831 ± 524.4, *p* = 0.030).

Table 2 shows the proportion of children with excess adiposity by each of the three anthropometric indexes among those with cardio-metabolic risk factors. It is worth noting that all children (*n* = 12) with high LDL-c had a WHtR ≥ 0.5.

Table 3 shows the areas under the ROC curves with 95% CI for each adiposity indicator tested for each cardio-metabolic risk marker. The AURC for high TC was statistically significant only for WHtR, while those for high LDL-c, low HDL-c, high TG, and high AIP were significant for all the three adiposity indicators, meaning that all of these anthropometric indexes were able to distinguish children with each lipidic risk factor from those without them. The WHtR showed the largest AURC for high LDL-c (0.742), low HDL-c (0.733), high TGs (0.734), and high TC (0.689), while z-BMI displayed the largest AURC for high AIP (0.831). It is important to highlight that AURC for high FPG and high AI were not statistically significant for any of the three adiposity indicators. When the AURC for each cardio-metabolic risk marker was compared among the anthropometric indicators, no statistical significance was found, except for FPG, although none of the AURCs for FPG was statistically significant, as previously described.

Estimates from linear regression models are shown in Table 4. The three adiposity indicators (predictors) were positively associated with LDL-c, TGs, AIP, and AI, and inversely associated with HDL-c in both univariate and adjusted models. All the three adiposity indexes were stronger associated with the AI and TG. The adjusted regression coefficients (β) showed that per each 0.1 unit increase in WHtR, TGs increased 0.34 mmol/L (95% CI: 0.22, 0.45; *p* = 0.000), and per 1 unit increase in WC and z-BMI, TGs increased 0.02 mmol/L (95% CI: 0.02, 0.03; *p* = 0.000) and 0.21 mmol/L (95% CI: 0.14, 0.29; *p* = 0.000), respectively.

Table 5 shows the ORs for the associations of excess general (z-BMI > 1 SD) and abdominal (WC ≥ 90 percentile or WHtR ≥ 0.5) adiposity with cardio-metabolic risk factors defined according to the criteria previously specified. All the adiposity indicators exhibited a significant association with low HDL and high TG, AIP and AI in both the univariate and multivariate models. Overall, higher raw and adjusted ORs were seen for the association between WHtR ≥ 0.5 and elevated TG. Those with a WHtR ≥ 0.5 had a 17.36-fold increased risk of having elevated TGs compared to those with a WHtR < 0.5, adjusted for energy intake (kcal), saturated fat (g) and carbohydrates (g), age and gender. Additionally, LDL-c was significantly associated only with WHtR when the Firth correction was applied; an increased risk (OR: 2.82; 95% CI: 0.75, 7.68; *p* = 0.003) of having LDL ≥ 3.4 mmol/L was observed among children with WHtR ≥ 0.5 compared to those with WHtR <0.5.

## 4. Discussion

Results of our study confirmed that WHtR performed similarly to WC and z-BMI in predicting lipidic cardio-metabolic risk factors in this sample of school-aged Mexican children, as all the three adiposity indicators showed significant AURCs greater than 0.68 for high LDL-c, low HDL-c, high TG, and high AIP. Additionally, a cut-off of 0.5 for WHtR was useful to detect significant increased risk of having lipidic cardio-metabolic risk factors, similar to that observed for a z-BMI > 1 SD and a WC ≥ 90 percentile. However, a WHtR ≥ 0.5 showed to be superior in detecting a significant increased risk of elevated LDL-c, even after adjustment for key variables. Importantly, none of the three adiposity indicators were able to discriminate high FPG in this study population.

The high prevalence of childhood obesity worldwide has guided us to search for alternatives to measure and predict the cardiovascular risk entailed in a better and faster way. Central obesity has a stronger association with an adverse cardio-metabolic risk profile in children either in the short and long term, compared to general obesity as determined by whole body fat [37,38,39,40]. Therefore, evaluation of abdominal obesity should be given sufficient importance in routine clinical practice during well child visits to improve the process of care. The use of a tool that is simple to measure and interpret as WHtR represents an advantageous alternative to screen for lipidic cardio-metabolic risk factors not only in the clinical setting, but also at the population level.

Diverse imaging techniques have been considered more accurate to evaluate fat content, such as bioimpedance analysis, dual energy X-ray absorptiometry (DEXA), air-displacement plethysmography (ADP), magnetic resonance imaging, etc. [41,42]; however, these are expensive methods, requiring more time and trained personal, and some of them imply radiation exposure. Thus, the utility of alternative easy to use and low-cost anthropometric indexes such as z-BMI, WC, and WHtR in assessing adiposity has been extensively evaluated. A recent systematic review of studies in children aged between 7 and 10 years old showed that BMI and WC were strongly correlated to body fat as measured by bioelectrical impedance or skinfolds, and that there was a moderate positive correlation with percent body fat as calculated by DEXA, ADP, or isotope dilution. However, in the case of WHtR, only a moderate positive correlation with body fat, as estimated by ADP and skinfolds, was reported [43]. In contrast, a study in children aged 8 to 18 years old concluded that WHtR was better than WC and BMI at predicting adiposity in this pediatric population, since it explained 80% of percent body fat variance, accounting for age and gender, as compared with 72% for WC and 68% for BMI [44]; in addition, a systematic review and meta-analysis reported that body fat measured by DEXA was strongly correlated with both BMI and WHtR, thus highlighting the utility of these two indexes to define obesity when more sophisticated techniques are not available [45]. Nonetheless, the relevance of the WHtR does not only rely on its ability to predict total body fat, but mainly on its potential superiority as a more practical tool to assess abdominal adiposity and to screen for cardio-metabolic risk [16,46].

In this regard, a meta-analysis of 34 studies in pediatric populations [21], including two in Mexican children and adolescents [23,24], reported that WHtR had significantly better screening power for elevated TGs compared to BMI and for high metabolic risk score compared to WC. Although WHtR was not superior for all the outcomes studied, authors highlighted its practical advantage in terms of measurement and interpretation for the screening of cardio-metabolic risk factors in children. The results of this present study extend the existing evidence base by reporting a similar good performance of the WHtR, WC, and BMI in the ROC analysis to discriminate children with lipidic cardio-metabolic risk factors from those with normal values, since all the three adiposity indicators exhibited significant AURCs greater than 0.68 for high LDL-c, low HDL-c, high TG, and high AIP; however, WHtR was the only one showing a significant AURC for high TC (0.689). Some other recent studies not included in the previous meta-analysis have also shown a similar performance of the WHtR compared to BMI or WC in identifying cardio-metabolic risk factors in pediatric population [47,48]. Importantly, in our study neither the AURCs for elevated FPG, nor the linear or logistic models constructed for this metabolic marker, were statistically significant for any of the three adiposity indicators. Similar findings were previously reported by other authors. In the meta-analysis by Lo et al. [21], two of three studies evaluating hyperglycemia reported no significant AURC for WHtR, WC, and BMI, and although the pooled AURC values for all indexes were statistically significant, they were just above 0.5 (0.57). In a more recent study among Korean children and adolescents, these same three adiposity indicators did not show a significant AURC for high fasting plasma glucose (FPG), defined as FPG ≥ 6.1 mmol/L, when stratified by gender [48]. Thus, it may be possible that other non-adiposity factors have a greater influence on glycemic status in pediatrics, such as genetics, intrauterine exposure, and environmental and lifestyle-related factors [49,50], which need to be further studied in order to address this metabolic alteration in early life.

It has been largely suggested that a cut-off point of 0.5 for WHtR is an effective indicator for health risks associated to obesity regardless of sex, age and ethnicity [19,51] and a recent meta-analysis concluded that this value is an appropriate cut-off for classifying cardio-metabolic risk in children and adolescents [21]. Nonetheless, diverse studies in children have suggested different cut-off points due to a higher predictive efficacy as compared to the traditional cut-off of 0.5 [16,23,24,48,52]. In Mexico, a cross-sectional study evaluating the utility of WHtR to predict metabolic syndrome in children between 6 and 12 years old, identified a value of 0.59 for WHtR as a strong predictor of this condition, whereas a cut-off of 0.5 showed very poor specificity (22.7%); we need to consider that most of the children (82%) in this study were overweight or obese, thus limiting the generalizability of the data. [23]. Similarly, another study in Mexican obese adolescents suggested that a WHtR ≥ 0.6 is a better predictor for metabolic syndrome compared to WC or BMI [24]. In our study, we tested the utility of the traditional cut-off for WHtR of 0.5 to evaluate the risk of presenting cardio-metabolic risk factors by logistic regression models adjusted by key dietary variables, as well as age and gender when applicable, since all these covariates were thought to influence such association. Our results showed that all the adiposity indicators had a significant association with low HDL and high TG, AIP, and AI in both the univariate and multivariate models; however, WHtR was the only of these indicators associated with LDL-C, showing a significant increased risk of elevated LDL-c (≥3.4 mmol/L) among children with WHtR ≥ 0.5 (OR: 2.82; 95% CI: 0.75, 7.68; *p* = 0.003) compared to those with a WHtR < 0.5, after adjustment for total energy (kcal), saturated fat (g), age, and gender.

Pediatric obesity is a public health problem worldwide, with rates in Latin America being among the highest in the world. Considering this, over the last few years diverse strategies and regulations aimed to prevent childhood obesity have been implemented in the Latin America region [53,54]. Here, as in all prevention strategies, monitoring is essential in order to document progress; in this regard, an effective and practical method to screen for obesity and its associated cardio-metabolic complications, as WHtR, may be of relevance. Additionally, the practicality of this method, as it does not require tables or graphics to be interpreted, makes of this an ideal approach for auto-screening at the population level to detect excess abdominal adiposity and its potential metabolic risk, and thus raising awareness among families regarding the need of searching for medical care and making life-style changes to improve children weight status and prevent further complications.

This study has some relevant limitations. Firstly, the small sample size may have led to the complete separation of data observed for the association between WHtR ≥ 0.5 and high LDL-c and between WC ≥ 90 percentile and high AIP, thus hindering parameter estimation during binomial logistic regression analyses; nonetheless, the estimates were able to be computed by applying the Firth correction [35]. Secondly, this is a cross-sectional design, and therefore a causal relationship between adiposity indicators and cardio-metabolic risk factors was not proved. Further longitudinal studies are still needed for this purpose. Thirdly, there are no well-established cut-off points for AIP and AI for the pediatric population, so we employed those used for adult population. Establishing cut-offs for these lipidic indexes and testing their usefulness as cardiovascular risk markers in pediatric population is necessary. Finally, dietary intake was measured by a single 24-h recall, which does not reflect the day-to-day variation in food intake; however, this method has been proven as valid for assessing the dietary intake in children as young as 8 years old without parental assistance [55]. In our study, we conducted this assessment with parental assistant to improve accuracy and quality of data collected.

In conclusion, we confirmed that WHtR, WC, and z-BMI performed similarly in predicting lipidic cardio-metabolic risk factors in this sample of school-aged Mexican children, while a cut-off of 0.5 for WHtR showed to be superior in detecting a significant increased risk of elevated LDL-c, even after adjustment for key variables. Importantly, none of the three adiposity indicators were able to discriminate high FPG, suggesting that other non-adiposity factors may have a greater influence on glucose metabolism in this pediatric population. Due to the increasing prevalence of obesity in early life and the cardio-metabolic risk associated with this weight condition, it is highly recommended to consider the use of the WHtR as a practical tool for auto-screening for abdominal obesity and cardio-metabolic risk at the population level for timely detection and management of increased cardio-metabolic risk in the pediatric population. 

## Figures and Tables

**Table 1 nutrients-10-01850-t001:** Characteristics of the sample by gender ^1^.

Characteristic	All	Boys (*n* = 52)	Girls (*n* = 73)	*p* ^2^
Age (years)	9 (8–10)	9 (8–10)	9 (8–10)	0.564
z-BMI (SD)	1.43 ± 1.19	1.52 ± 1.24	1.35 ± 1.15	0.407
Waist circumference (cm)	73.42 ± 12.79	72.95 ± 13.62	73.76 ± 12.25	0.726
Waist-to-height ratio	0.52 ± 0.07	0.52 ± 0.07	0.52 ± 0.07	0.803
Systolic BP (mmHg)	96.83 ± 8.99	97.51 ± 8.88	96.35 ±9.10	0.481
Diastolic BP (mmHg)	64.13 ± 6.68	64.19 ± 7.80	64.09 ± 5.8	0.934
Fasting plasma glucose (mmol/L)	5.05 (4.89–5.22)	5.05 (4.90–5.22)	5.05 (4.83–5.22)	0.534
Total cholesterol (mmol/L)	4.21 ± 0.68	4.24 ± 0.67	4.20 ± 0.69	0.745
LDL cholesterol (mmol/L)	2.66 ± 0.64	2.65 ± 0.62	2.66 ± 0.65	0.927
HDL cholesterol (mmol/L)	1.27 (1.12–1.54)	1.30 (1.13–1.59)	1.24 (1.10–1.49)	0.200
Triglycerides (mmol/L)	1.01 (0.74–1.40)	1.01 (0.69–1.35)	1.01 (0.79–1.49)	0.321
AI	1.98 (1.61–2.44)	1.95 (1.59–2.39)	2.01 (1.61–2.59)	0.595
AIP	0.25 ± 0.26	0.21 ± 0.26	0.28 ± 0.26	0.163
Energy intake (kcal/day)	1935 ± 610.2	2081 ± 692.6	1831 ± 524.4	0.030
Protein (g/day)	76.95 ± 26.07	78.44 ± 28.5	75.88 ± 24.31	0.590
Total fat (g/day)	62.9 (48.1–82.5)	63.13 (49.81–105.93)	62.9 (47.5–81.0)	0.256
Saturated fat (g/day)	23.8 (15.3–30.18)	25.58 (15.9–31.8)	22.5 (14.7–29.8)	0.348
Carbohydrates (g/day)	236.9 (186–319)	260.14 (191–346)	231.61 (181–294)	0.564

SD: standard deviation; z-BMI: z-score body mass index; BP: blood pressure; LDL: low-density lipoprotein; HDL: high-density lipoprotein; AI: atherogenic index; AIP: atherogenic index of plasma. ^1^ Values are mean ± standard deviation or median (25th–75th percentiles). ^2^ For comparison between boys vs. girls by using Student’s *t*-test for independent samples or Mann–Whitney U test.

**Table 2 nutrients-10-01850-t002:** Proportion of children with obesity by the three adiposity indicators among those with cardio-metabolic risk factors.

Cardio-Metabolic Risk Marker	WHtR ≥ 0.5 *n* (%)	WC ≥ 90 Percentile *n* (%)	BMI z-Score > 1 SD *n* (%)
LDL-c ≥ 3.4 mmol/L (*n* = 12)	12 (100)	7 (58.3)	11 (91.6)
HDL-c < 1 mmol/L (*n* = 15)	14 (93.3)	10 (66.6)	14 (93.3)
TGs ≥1.1 mmol/L (0–9 years) or ≥ 1.5 mmol/L (10–19 years) (*n* = 40)	37 (92.5)	22 (55)	34 (85)
FPG ≥ 5.6 mmol/L (*n* = 6)	3 (50)	3 (50)	3 (50)
TC ≥ 5.2 mmol/L (*n* = 11)	10 (90.9)	5 (45.45)	9 (81.8)
AIP > 0.11 (*n* = 92)	70 (76)	44 (47.8)	71 (77.1)
AI > 3 for women and > 2.5 for men (*n* = 27)	26 (96.2)	17 (62.9)	25 (92.6)

WHtR: waist-to-height ratio; WC: waist circumference; z-BMI: z-score body mass index; CI: confidence interval; LDL: low-density lipoprotein; HDL: high-density lipoprotein; TGs: triglycerides; TC: total cholesterol; AI: atherogenic index; AIP: atherogenic index of plasma.

**Table 3 nutrients-10-01850-t003:** Association of each adiposity indicator and cardio-metabolic risk factors using receiver operating characteristic curves.

Cardio-Metabolic Risk Marker	WHtR	WC	z-BMI	
	AURC (95% CI) *p* Value	AURC (95% CI) *p* Value	AURC (95% CI) *p* Value	*p* *
LDL-c ≥ 3.4 mmol/L	0.742 (0.63, 0.86) 0.006	0.685 (0.54, 0.83) 0.035	0.687 (0.55, 0.82) 0.034	0.1511
HDL-c < 1 mmol/L	0.733 (0.60, 0.87) 0.004	0.719 (0.59, 0.85) 0.006	0.704 (0.57, 0.83) 0.011	0.8124
TGs ≥ 1.1 mmol/L (0–9 years) or ≥1.5 mmol/L (10–19 years)	0.734 (0.65, 0.82) 0.000	0.690 (0.59, 0.79) 0.001	0.730 (0.64, 0.82) 0.000	0.2382
FPG ≥ 5.6 mmol/L	0.507 (0.21, 0.80) 0.954	0.627 (0.32, 0.93) 0.293	0.574 (0.29, 0.86) 0.544	0.003
TC ≥ 5.2 mmol/L	0.689 (0.55, 0.83) 0.039	0.630 (0.47, 0.80) 0.155	0.655 (0.50, 0.81) 0.090	0.3034
AIP > 0.11	0.811 (0.73, 0.89) 0.000	0.825 (0.75, 0.90) 0.000	0.831 (0.75, 0.91) 0.000	0.7050
AI > 3 for women and >2.5 for men	0.699 (0.46, 0.94) 0.335	0.681 (0.56, 0.80) 0.381	0.652 (0.39, 0.92) 0.461	0.2392

WHtR: waist-to-height ratio; WC: waist circumference; z-BMI: z-score body mass index; CI: confidence interval; LDL: low-density lipoprotein; HDL: high-density lipoprotein; TGs: triglycerides; TC: total cholesterol; AI: atherogenic index; AIP: atherogenic index of plasma. * *p* value for comparison among ROC curves.

**Table 4 nutrients-10-01850-t004:** Univariate and multivariate linear association of adiposity indicators and cardio-metabolic risk factors.

Cardio-Metabolic Risk Marker	WHtR (Per Increase of 0.1 Units) *				WC (cm) *				z-BMI (SD)			
	β	95% CI	Standardized β	*p*	β	95% CI	Standardized β	*p*	β	95% CI	Standardized β	*p*
LDL-c (mmol/L) Unadjusted model	0.27	0.12, 0.41	0.31	0.000	0.01	0.00, 0.02	0.24	0.007	0.13	0.03, 0.21	0.24	0.008
Adjusted model ^1^	0.24	0.09, 0.38	0.28	0.002	0.01	0.00, 0.02	0.29	0.005	0.12	0.02, 0.21	0.22	0.015
HDL-c (mmol/L) Unadjusted model	−0.21	−0.27, −0.14	−0.49	0.000	−0.01	−0.02, −0.01	−0.53	0.000	−0.12	−0.16, −0.08	−0.45	0.000
Adjusted model ^2^	−0.21	−0.27, −0.14	−0.49	0.000	−0.01	−0.02, −0.01	−0.56	0.000	−0.12	−0.17, −0.08	−0.47	0.000
TGs (mmol/L) Unadjusted model	0.34	0.23, 0.46	0.47	0.000	0.02	0.02, 0.03	0.54	0.000	0.20	0.13, 0.27	0.45	0.000
Adjusted model ^2^	0.34	0.22, 0.45	0.47	0.000	0.02	0.02, 0.03	0.55	0.000	0.21	0.14, 0.29	0.48	0.000
TC (mmol/L) Unadjusted model	0.13	−0.03, 0.29	0.14	0.108	0.00	−0.01, 0.01	0.07	0.424	0.06	−0.05, 0.16	0.10	0.277
Adjusted model ^1^	0.10	−0.06, 0.27	0.11	0.217	0.01	−0.01, 0.02	0.11	0.299	0.05	−0.06, 0.15	0.08	0.386
FPG (mmol/L) Unadjusted model	0.03	−0.04, 0.10	0.07	0.421	0.00	−0.00, 0.01	0.15	0.089	0.03	−0.02, 0.07	0.11	0.237
Adjusted model ^3^	0.02	−0.06, 0.10	0.05	0.597	0.00	−0.00, 0.01	0.14	0.119	0.02	−0.02, 0.07	0.09	0.328
AIP Unadjusted model	0.20	0.14, 0.25	0.54	0.000	0.01	0.01, 0.016	0.62	0.000	0.12	0.08, 0.15	0.52	0.000
Adjusted model ^4^	0.20	0.14, 0.25	0.55	0.000	0.01	0.01, 0.016	0.63	0.000	0.12	0.09, 0.16	0.55	0.000
AI Unadjusted model	0.47	0.33, 0.63	0.49	0.000	0.03	0.02, 0.036	0.47	0.000	0.25	0.15, 0.35	0.41	0.000
Adjusted model ^4^	0.48	0.33, 0.64	0.50	0.000	0.03	0.02, 0.035	0.46	0.000	0.26	0.16, 0.36	0.43	0.000

WHtR: waist-to-height ratio; WC: waist circumference; z-BMI: z-score body mass index; SD: standard deviation; CI: confidence interval; LDL: low-density lipoprotein; HDL: high-density lipoprotein; TGs: triglycerides; TC: total cholesterol; AIP: atherogenic index of plasma; AI: atherogenic index. ^1^ Adjusted for total energy (kcal) and saturated fat (g); ^2^ Adjusted for total energy (kcal), saturated fat (g) and carbohydrates (g); ^3^ Adjusted for total energy (kcal) and carbohydrates (g); ^4^ Adjusted for total energy (kcal), total fat (g) and carbohydrates (g). * All multivariate models for WHtR and WC were further adjusted for age and gender.

**Table 5 nutrients-10-01850-t005:** Logistic regression analyses for the association of excess general (z-BMI > 1 SD) and abdominal (WC ≥ 90 percentile or WHtR ≥ 0.5) adiposity and cardio-metabolic risk factors.

Cardio-Metabolic Risk Marker	WHtR ≥ 0.5 *			WC ≥ 90 Percentile			BMI z-Score > 1 SD		
	OR	95% CI	*p*	OR	95% CI	*p*	OR	95% CI	*p*
LDL-c ≥ 3.4 mmol/L Unadjusted model	2.91 ^¥^	0.84, 7.78	0.002	3.12	0.93, 10.51	0.066	7.01	0.86, 56.25	0.067
Adjusted model ^1^	2.82 ^¥^	0.75, 7.68	0.003	3.39	0.97, 11.81	0.055	6.47	0.79, 52.83	0.081
HDL-c < 1 mmol/L Unadjusted model	11.02	1.41, 86.34	0.022	4.88	1.54, 15.39	0.007	9.33	1.18, 73.54	0.034
Adjusted model ^2^	11.05	1.36, 89.77	0.025	5.20	1.60, 16.90	0.006	11.06	1.37, 89.43	0.024
TGs ≥ 1.1 mmol/L for 0–9 years and ≥ 1.5 mmol/L for 10–19 years Unadjusted model	13.90	3.99, 48.49	0.000	3.97	1.79, 8.84	0.001	4.80	1.83, 12.64	0.001
Adjusted model ^2^	17.36	4.65, 64.77	0.000	4.44	1.93, 10.23	0.000	5.32	1.94, 14.58	0.001
TC ≥ 5.2 mmol/L Unadjusted model	7.60	0.95, 60.89	0.056	2.60	0.74, 9.08	0.134	2.73	0.56, 13.21	0.213
Adjusted model ^1^	6.83	0.82, 56.89	0.076	2.67	0.76, 9.41	0.126	2.54	0.51, 12.55	0.253
FPG ≥ 5.6 mmol/L Unadjusted model	0.59	0.12, 3.06	0.532	2.05	0.40, 10.63	0.392	0.55	0.11, 2.82	0.470
Adjusted model ^3^	0.52	0.09, 3.00	0.464	2.07	0.40, 10.77	0.388	0.47	0.09, 2.57	0.383
AIP ≥ 0.11 Unadjusted model	12.16	4.65, 31.79	0.000	4.00 ^¥^	2.00, 8.85	0.000	9.02	3.64, 22.35	0.000
Adjusted model ^4^	14.16	4.86, 41.25	0.000	4.16 ^¥^	2.14, 9.02	0.000	10.22	3.85, 27.11	0.000
AI > 3 for women and < 2.5 for men Unadjusted model	11.06	2.48, 49.26	0.002	4.96	2.00, 12.25	0.001	9.77	2.19, 43.56	0.003
Adjusted model ^4^	12.39	2.71, 56.65	0.001	5.11	2.02, 12.94	0.001	10.78	2.33, 49.78	0.002

WHtR: waist-to-height ratio; WC: waist circumference; z-BMI: z-score body mass index; CI: confidence interval; LDL: low-density lipoproteins; HDL: high-density lipoproteins; TGs: triglycerides; TC: total cholesterol; AIP: atherogenic index of plasma; AI: atherogenic index. ^1^ Adjusted for total energy (kcal) and saturated fat (g); ^2^ Adjusted for total energy (kcal), saturated fat (g) and carbohydrates (g); ^3^ Adjusted for total energy (kcal) and carbohydrates (g); ^4^ Adjusted for total energy (kcal), total fat (g) and carbohydrates (g). ^¥^ Estimated by the penalized Firth logistic regression method. * All adjusted models for WHtR were further adjusted for age and gender.

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
