# Peer review of "Performance of Waist-To-Height Ratio, Waist Circumference, and Body Mass Index in Discriminating Cardio-Metabolic Risk Factors in a Sample of School-Aged Mexican Children"

_nutrients, 2018, doi:10.3390/nu10121850_

Round 1

Reviewer 1 Report

The study aimed to determine utility of three anthropometric indices – waist-to-height ratio (WHtR), waist circumference (WC), and body mass index (BMI) to screen lipidic and non-lipidic cardio-metabolic risk factors in Mexican children aged between 8 – 10 years. The results showed that, while all three indices associated with atherogenic indices and lipid profiles, WHtR was significantly associated with high LDL. Fasting plasma glucose was however, not associated with all anthropometric indices.

The manuscript is easy to follow and explicit throughout. Also it is clear that the study was conducted in appropriate manner with regards to informed consent from both children and their caregivers. It was also good that the authors acknowledged limitations of the study and stated in the manuscript. The manuscript will improve its clarity if authors consider following points:

-        While the authors stated that 125 children aged 6 – 12 years were analyzed in the abstract, the results showed the participants were 125 children aged 8 – 10. Please clarify and make a consistent statement.

-        A valid and reliable anthropometric measurement is a key component of the current study. However it is unclear whether all measurements of the current study was conducted by the same anthropometrist or the data was collected by a group of researchers. Also it is not clear if the measurer was well trained or experienced. It is recommended to include a level of technical error of measurement in the manuscript.

-        While a blood sample was obtained from participants, it is not clear how much in volume the participants were requested to provide. Since it is important to understand a level of invasiveness to the readers, it is recommended the authors to include in the manuscript.

Author Response

Dear reviewer:

We appreciate your comments,  we have addressed all of them.

Please see the point-by-point response to each comment.

Response to reviewer

The study aimed to determine utility of three anthropometric indices – waist-to-height ratio (WHtR), waist circumference (WC), and body mass index (BMI) to screen lipidic and non-lipidic cardio-metabolic risk factors in Mexican children aged between 8 – 10 years. The results showed that, while all three indices associated with atherogenic indices and lipid profiles, WHtR was significantly associated with high LDL. Fasting plasma glucose was however, not associated with all anthropometric indices.

The manuscript is easy to follow and explicit throughout. Also it is clear that the study was conducted in appropriate manner with regards to informed consent from both children and their caregivers. It was also good that the authors acknowledged limitations of the study and stated in the manuscript. The manuscript will improve its clarity if authors consider following points:

-        While the authors stated that 125 children aged 6 – 12 years were analyzed in the abstract, the results showed the participants were 125 children aged 8 – 10. Please clarify and make a consistent statement.

Enrolled children were 6 to 12 years old, as stated in the ‘Abstract’: “Overall, 125 children aged 6 to 12 years were analyzed…”

It is also stated in the ‘2.1 Study population’ section: “This cross-sectional analysis included 125 healthy children between 6 to 12 years old…”

In the ‘Results’ section, 9 (8-10) refers to the median (25th-75th percentiles). It is stated as follows: “The median (25th-75th percentiles) age was 9 (8-10) years old and…”

-        A valid and reliable anthropometric measurement is a key component of the current study. However it is unclear whether all measurements of the current study was conducted by the same anthropometrist or the data was collected by a group of researchers. Also it is not clear if the measurer was well trained or experienced. It is recommended to include a level of technical error of measurement in the manuscript.

Anthropometric measures were taken by the same previously standardized research personnel. Standardization procedures were carried out according to the Habicht method.

It was clarified in the ‘2.2.1 Anthropometry and adiposity indicators definition’ section: Weight, height and waist circumference were measured following standardized procedures described by the International Society for the Advancement of Kinanthropometry (ISAK)[25]. All anthropometric measures were taken by the same previously standardized research personnel. Standardization procedures were carried out according to the Habicht method (26).”

Reference number 26 was added.

-       While a blood sample was obtained from participants, it is not clear how much in volume the participants were requested to provide. Since it is important to understand a level of invasiveness to the readers, it is recommended the authors to include in the manuscript.

Children were requested to provide a total of 16 mL, and 3.5 mL were processed for the purpose of the determinations used in this analysis. It was clarified in the ‘2.2.2. Biochemical analysis and cardio-metabolic risk factors’ section, it reads as follows: “Fasting blood samples (16 ml) were collected. For the purpose of the biochemical determinations employed in this sub-analysis, 3.5 ml were processed by the central laboratory of the INCICh to determine serum levels of...”

Reviewer 2 Report

This is a cross sectional study in Mexican children to assess association between three anthropometric measurements (WHtR, WC and BMI) and different cardio-metabolic risk factors; and their prediction capability.  

Major comments

-          The title does not reflect the results of the study. 1) no proper discrimination analysis was undertaken; and 2) no formal comparison between the three measures to conclude about ‘superiority”.

-          Lines 19-21, abstract. “The objective of this study was to evaluate the utility of WHtR, WC and body mass index to detect …” The methodologic approach does not assess “utility”. The study has assessed “association” (linear and logistic analyses) and prediction capability of each measure (ROC analysis). Same comment applies for Lines 61-64.

-          According to lines 67-70 this is a secondary analysis of another study aimed at assessing “echocardiography abnormalities associated with obesity in school age children”. From this, I understand that the target population were obese children. Please describe in detail the inclusion criteria used to select the children (based on BMI reported in Table 1 it seems that most of the children were overweight or obese).

-          Dietary intake. Was the FFQ self-reported or taken by a trained interviewer?

-          Please include a table with the number and percentage of children in each of the levels of cardiovascular markers (column 1 table 2) and of the anthropometric measures dichotomized as in in Table 4. This information is key to

o   understand the large CI associated with some of the AUROC estimates;

o   potentially explain the separation issue described in the statistical analysis; and  

o   add to the description in lines 188-197 and Table 4.

-          Table 2. Please include tests for the comparison between AUROC estimated under the three models.

-          Line 208-214. This summary of results does not cover all the important results in the paper. Consider moving the summary in the conclusion paragraph to the beginning of the discussion.

Minor comments

-          Table 1 – Line 159- Should be “independent” instead of “interdependent”?

-          Paragraph relative to Table 2, explain what “significant” means for an ROC analysis.

-          Line 172 – “Linear regression models are shown …” suggest changing by “Estimates from linear regression models are shown…”

Author Response

Dear reviewer:

We appreciate your comments, we have addressed all of them.

Please see the point-by-point response to each comment.

Response to rerviewer

This is a cross sectional study in Mexican children to assess association between three anthropometric measurements (WHtR, WC and BMI) and different cardio-metabolic risk factors; and their prediction capability. 

Major comments

-          The title does not reflect the results of the study. 1) no proper discrimination analysis was undertaken; and 2) no formal comparison between the three measures to conclude about ‘superiority”.

We agree. Although WHtR was the only adiposity indicator associated with an increased risk of elevated LDL, and numerically AURC was greater for this adiposity indicator, no clear superiority was demonstrated. It was further confirmed when the AURC were compared, as described below in the following responses. Thus, we modified the title and now it reads as follows: “Performance of waist-to-height ratio, waist circumference and body mass index in discriminating cardio-metabolic risk factors in a sample of Mexican school age children”.

-          Lines 19-21, abstract. “The objective of this study was to evaluate the utility of WHtR, WC and body mass index to detect …” The methodologic approach does not assess “utility”. The study has assessed “association” (linear and logistic analyses) and prediction capability of each measure (ROC analysis).

Done, it was modified as follows: “The objective of this study was to evaluate the association between WHtR, WC and body mass index with lipidic and non-lipidic cardio-metabolic risk factors and the prediction capability of each adiposity indicator in a sample of Mexican school-age children.”

-         Same comment applies for Lines 61-64.

Done, it was also modified. It now reads as follows: “The aim of this study was to evaluate the association of WHtR and WC as indicators of abdominal adiposity, and BMI as an indicator of general adiposity, with lipidic and non-lipidic cardio-metabolic risk factors and their prediction capability, in a sample of Mexican school age children, considering the effect of key dietary indicators.”

-          According to lines 67-70 this is a secondary analysis of another study aimed at assessing “echocardiography abnormalities associated with obesity in school age children”. From this, I understand that the target population were obese children. Please describe in detail the inclusion criteria used to select the children (based on BMI reported in Table 1 it seems that most of the children were overweight or obese).

Children with normal weight, overweight and obesity were recruited. Only those with underweight were excluded. It was clarified in the ‘2.1. Study Population’ section as follows: “Children with underweight, cardiovascular disease, hypertension, diabetes or any other chronic condition were excluded of the study.”

Additionally, in the ‘2.2.1. Anthropometry and adiposity indicators definition’ section, it is stated that children were classified as having normal weight, overweight and obesity, according to WHO criteria for z-BMI. It is stated as follows: “BMI z-score was calculated, as an indicator of general adiposity, and classified according to the World Health Organization (WHO) age- and gender-specific growth standards with the WHO Anthro Plus software [26]. Normal weight was defined as a BMI z-score between −2 standard deviations (SD) and 1 SD; overweight, z-score >1 SD; and obesity, z-score >2 DE [27].”                                       

-          Dietary intake. Was the FFQ self-reported or taken by a trained interviewer?

It was taken by a trained research dietitian. It was clarified in the ‘2.2.3. Dietary intake’ section.

-          Please include a table with the number and percentage of children in each of the levels of cardiovascular markers (column 1 table 2) and of the anthropometric measures dichotomized as in in Table 4. This information is key to

o   understand the large CI associated with some of the AUROC estimates;

o   potentially explain the separation issue described in the statistical analysis; and  

o   add to the description in lines 188-197 and Table 4.

Table 2 was included. It shows that all children (n=12) with high LDL-c had a WHtR ≥0.5.

It is referenced in the ´Results´ sections as follows: “Table 2 shows the proportion of children with excess adiposity by each of the three anthropometric indexes among those with cardio-metabolic risk factors. It is worth noting that all children (n=12) with high LDL-c had a WHtR ≥0.5.”

-          Table 2. Please include tests for the comparison between AUROC estimated under the three models.

It was included in Table 3 (former table 2). There were no significant differences among AURC of each adiposity indicator for any of the cardio-metabolic risk marker, except for FPG; however, ROC curves for FPG were not statistically significant for any of the three adiposity indicators.

These comparisons were declared in the ‘2.3. Statistical analysis’ section, it reads as follows: “ROC curves for each cardio-metabolic risk factor were compared among the three adiposity indicators…” and “All analyses were performed using SPSS Version 23.0 (SPSS, Inc., Chicago, IL, USA), except for the comparison of the AURC among the three adiposity indicators, which was performed applying the roccomp command in Stata v.14…”

These results were described in the ‘3. Results’ section as follows: “…When the AURC for each cardio-metabolic risk marker were compared among the anthropometric indicators, no statistical significance was found, except for FPG, although none of the AURC for FPG was statistical significant as previously described…

-          Line 208-214. This summary of results does not cover all the important results in the paper. Consider moving the summary in the conclusion paragraph to the beginning of the discussion.

Done. Summary in the conclusions was moved to the beginning of the discussion and the conclusions were modified in order not to read exactly the same. Changes read as follows:

a)    Discussion: “Results of our study confirmed that WHtR performed similarly to WC and z-BMI in predicting lipidic cardio-metabolic risk factors in this sample of Mexican school-age children, as all the three adiposity indicators showed significant AURCs greater than 0.68 for high LDL-c, low HDL-c, high TG, and high AIP. Additionally, a cut-off of 0.5 for WHtR was useful to detect significant increased risk of having lipidic cardio-metabolic risk factors, similar to that observed for a z-BMI >1SD and a WC ≥90 percentile; however, a WHtR ≥0.5 showed to be superior in detecting a significant increased risk of elevated LDL-c, even after adjustment for key variables. Importantly, none of the three adiposity indicators were able to discriminate high FPG in this study population.”

b)    Conclusions: “In conclusion, we confirmed that WHtR, WC, and z-BMI performed similarly in predicting lipidic cardio-metabolic risk factors in this sample of Mexican school-age children, while a cut-off of 0.5 for WHtR showed to be superior in detecting a significant increased risk of elevated LDL-c, even after adjustment for key variables. Importantly, none of the three adiposity indicators were able to discriminate high FPG, suggesting that other non-adiposity factors may have a greater influence on glucose metabolism in this pediatric population. Due to the increasing prevalence of obesity in the early life and the cardio-metabolic risk associated with this weight condition, it is highly recommended to consider the use of the WHtR as a practical tool for auto-screening for abdominal obesity and cardio-metabolic risk at the population level for a timely detection and management of increased cardio-metabolic risk in pediatric population.” 

Minor comments

-          Table 1 – Line 159- Should be “independent” instead of “interdependent”?

You are right, it was modified.

-          Paragraph relative to Table 2, explain what “significant” means for an ROC analysis.

It was stated as follows: “The AURC for high TC was statistically significant only for WHtR, while those for high LDL-c, low HDL-c, high TG, and high AIP were significant for all the three adiposity indicators; meaning that all of these anthropometric indexes were able to distinguish children with each lipidic risk factors from those without them.”

-          Line 172 – “Linear regression models are shown …” suggest changing by “Estimates from linear regression models are shown…”

Done, it was modified.